# The Effects of 21-Day General Rehabilitation after Hip or Knee Surgical Implantation on Plasma Levels of Selected Interleukins, VEGF, TNF-α, PDGF-BB, and Eotaxin-1

**DOI:** 10.3390/biom12050605

**Published:** 2022-04-19

**Authors:** Maciej Idzik, Jakub Poloczek, Bronisława Skrzep-Poloczek, Ewelina Dróżdż, Elżbieta Chełmecka, Zenon Czuba, Jerzy Jochem, Dominika Stygar

**Affiliations:** 1Independent Public Health Care, Opole Cancer Center Prof. Tadeusz Koszarowski, Katowicka 45-061 Street, 46-020 Opole, Poland; macidzik@tlen.pl; 2Department of Rehabilitation, 3rd Specialist Hospital in Rybnik, Energetyków 46 Street, 44-200 Rybnik, Poland; poloczek.jakub@gmail.com; 3Department of Internal Medicine, Diabetology and Nephrology, Faculty of Medical Sciences in Zabrze, Medical University of Silesia, 40-055 Katowice, Poland; 4Department of Physiology, Faculty of Medical Sciences in Zabrze, Medical University of Silesia, Jordana Street 19, 40-055 Katowice, Poland; rehabilitacja-kopernik@o2.pl (B.S.-P.); jjochem@poczta.onet.pl (J.J.); 5Department of Statistics, Department of Instrumental Analysis, Faculty of Pharmaceutical Sciences in Sosnowiec, Medical University of Silesia, 40-055 Katowice, Poland; edrozdz@sum.edu.pl (E.D.); echelmecka@sum.edu.pl (E.C.); 6Department of Microbiology and Immunology, School of Medicine with the Division of Dentistry in Zabrze, Medical University of Silesia, 40-055 Katowice, Poland; zczuba@sum.edu.pl

**Keywords:** cytokines, chemokines, general rehabilitation, interleukins, hip or knee implantation surgery, osteoarthritis

## Abstract

Rehabilitation in osteoarthritis (OA) patients aims to reduce joint pain and stiffness, preserve or improve joint mobility, and improve patients’ quality of life. This study evaluated the effects of the 21-day individually adjusted general rehabilitation program in 36 OA patients 90 days after hip or knee replacement on selected interleukins (IL) and cytokines using the Bio-Plex^®^ Luminex^®^ system. Serum concentrations of almost all selected anti/pro-inflammatory markers: IL-1 receptor antagonist (IL-1RA), IL-2, IL-4, IL-5, IL-6, IL-10, IL-13, IL-15, and of some chemokines: macrophage inflammatory protein-1 alpha (MIP-1α/CCL3), and RANTES/CCL5, and of eotaxin-1/CCL11, the vascular endothelial growth factor (VEGF) significantly increased, whereas basic fibroblast growth factor (FGF basic) significantly decreased after the 21-day general rehabilitation. The levels of interferon-γ induced protein 10 (IP-10), MIP-1β/CCL4, macrophage/monocyte chemoattractant protein-1 (MCP-1/CCL2 (MCAF)), granulocyte macrophage-colony stimulating factor (GM-CSF), platelet-derived growth factor-BB (PDGF-BB), and granulocyte colony-stimulating factor (G-CSF) remained unchanged. There were no changes in pro-inflammatory cytokines levels: tumor necrosis factor-alpha (TNF-α), interferon-γ (IFN-γ), and IL-12 (p70)) after the 21-day general rehabilitation, indicating the stable and controlled inflammatory status of osteoarthritis patients. Significantly higher levels of anti-inflammatory factors after 21 days of moderate physical activity confirm the beneficial outcome of the applied therapy. The increased level of IL-6 after the rehabilitation may reflect its anti-inflammatory effect in osteoarthritis patients.

## 1. Introduction

Osteoarthritis (OA), as one of the most common chronic health conditions worldwide, is associated with a large societal and economic burden. It is a major cause of pain and disability that progress with age [1,2]. Cartilage destruction, subchondral bone remodeling, and synovial membrane inflammation are the main signs of OA [3]. Synovial membrane inflammation and several immune processes within the joint are strongly implicated in the pathogenesis and progression of this pathology, even though traditionally, OA was classified as a non-inflammatory arthropathy [4]. It is well established that the breakdown of cartilage extracellular matrix releases from chondrocytes and synovial cells various catabolic and pro-inflammatory mediators, which contributes to further destruction of cartilage. At the same time, the repair processes are also impaired [5]. Moreover, the molecular signals of tissue damage also trigger innate immune response [6], releasing inflammatory mediators, such as cytokines, that can also play a major role in the pathogenic processes within the joint [7,8]. The low-grade systemic inflammation could also initiate or aggravate OA, and, in turn, the locally-produced OA mediators may impact on the perpetuation of this systemic inflammation. Thus, the inflammation occurring within the joint tissues might be reflected outside the joint, i.e., in the peripheral blood of OA patients [9,10]. The low-grade systemic inflammation can also negatively affect the feedback between the inflammatory and oxidative stress responses in OA patients [11].

Over the last decade, a concept emerged in exercise biology that skeletal muscles and other organs initiate tissue-to-tissue crosstalk in response to exercise by the secretion and release of circulating factors [12]. These exercise-stimulated factors include proteins, peptides, hormones, metabolites, and cytokines. Cytokines regulate the human immune response by interacting with specific cytokine inhibitors and soluble cytokine receptors. They play a physiologic role in inflammation and a pathologic role in systemic inflammatory states [13]. Cytokines are 5–20 kDa proteins that play an essential role in cell and tissue-to-tissue signaling, and some of them are known to induce chemotaxis.

Chemokines (chemotactic cytokines), a subgroup of structurally related cytokines, are mainly responsible for leukocytes activation and migration. All chemokines have conserved cysteine residues that allow for the distinguishing of four groups: C-C chemokines (eotaxin-1/CCL11 (C–C motif ligand 11 chemokine), RANTES/CCL5, monocyte chemoattractant protein (MCP-1/CCL2), monocyte inflammatory protein-1 alpha and beta (MIP-1α/CCL3 and MIP-1β/CCL4)); C-X-C chemokines (interleukin 8 (IL-8)/growth-related oncogene (GRO/KC)); C chemokines (lymphotactin); and CXXXC chemokines (fractalkine) [13,14].

Pro-inflammatory and immunoregulatory interleukins (IL-1, IL-6), tumor necrosis factor-alpha (TNF-α), interferon gamma (IFN-γ), and granulocyte-macrophage colony-stimulating factor (GM-CSF) play a significant role in the pathogenesis of many autoinflammatory diseases. IL-1 and TNF-α, in particular, are connected to the joints’ destruction in rheumatoid arthritis. Low levels of IFN-γ and higher levels of GM-CSF, IL-2, and IL-6 were detected in rheumatoid arthritis synovial effusions [15]. Many cytokines stimulate IL-1 or TNF-α production by peripheral blood mononuclear cells and monocytes. Others, like lL-4, have an inhibitory effect [15].

The anti-inflammatory cytokines control the pro-inflammatory cytokines response. Major anti-inflammatory cytokines include interleukin IL-4, IL-10, and IL-13 [13]. IL-4 and IL-13 express pleiotropic effects depending on the environment, but most OA studies report their anti-inflammatory effect. IL-4 is associated with a strong chondroprotective effect. It inhibits the degradation of proteoglycans in the articular cartilage by reducing the secretion of metalloproteinases (MMPs) and the variation in the proteoglycans production observed in OA [16,17,18]. Moreover, IL-4 protects against post-traumatic osteoarthritis in mice and down-regulates osteoarthritis-associated genes in human synovial tissue [19].

IL-4 biological activity is mediated through a receptor system mutual to IL-4 and IL-13. The IL-13 anti-inflammatory effect in OA is related to synovium fibroblasts and is based on inhibiting the secretion of inflammatory cytokines released by macrophages, monocytes, B cells, NK cells, and endothelial cells [16,20]. IL-13 was reported to inhibit the synthesis of pro-inflammatory IL-1β, TNF-α, and MMP-3 and simultaneously increase IL-1RA in the synovial fluid [21]. Moreover, IL-13 inhibited the TNF-α pro-inflammatory effect in fibroblasts from OA patients [17].

IL-10 is a cytokine with potent anti-inflammatory properties. It represses the expression of inflammatory cytokines such as TNF-α, IL-6, and IL-1 produced by activated macrophages, synovial fibroblasts, or chondrocytes. Additionally, IL-10 up-regulates endogenous anti-cytokines and is a potent inhibitor of IL-1, TNF-α, and metalloproteases production, and also down-regulates pro-inflammatory cytokine receptors [20]. Thus, it counter-regulates the production and function of pro-inflammatory cytokines at multiple levels. Clinical studies indicate that low IL-10 and IL-4 blood levels could be responsible for widespread chronic pain [22].

IL-1RA (interleukin-1 receptor antagonist) is a ~17.6 kDa protein that binds competitively with interleukin-1 receptor type 1 (IL-1R1) and blocks cells’ activation by IL-1. Based on encouraging in vitro and pre-clinical in vivo data from experimental arthritis and osteoarthritis models, IL-1RA is considered a promising disease-modifying OA drug [23].

GM-CSF and IL-5 (β common chain cytokines) regulate various inflammatory responses leading to rapid pathogen removal and contribute to chronic pathological inflammation [24]. IL-6 is also categorized as either an anti-inflammatory or pro-inflammatory cytokine, depending on the circumstances [13].

The presented study analyzes the effects of a personalized 21-day general rehabilitation program on the selected interleukins and cytokines in OA patients to understand the systemic effects. The applied 21-day general rehabilitation aimed to reduce pain and stiffness of the joints, preserve and improve joint mobility and quality of patient’s life, and prevent or reduce the progression of the disease [11,25,26].

## 2. Materials and Methods

The study complied with the Declaration of Helsinki requirements. The study protocol was approved by the Ethics Committee of The Medical University of Silesia in Katowice (N° KNW/002/KB1/106/17; 3 October 2017). The participants were informed about the study’s positive and negative aspects (tiredness, possible additional joint and muscle pain, and time obligation), received a written description of the protocol, and returned the written consent to participate before the study started.

### 2.1. Study Group

A total of 62 candidate patients after hip or knee implantation in the course of osteoarthritis were available among patients of the outpatient clinic during the study (2017–2018). The candidate participants (>18 y.o. after hip or knee arthroplasty due to osteoarthritis in the prior 3 months) were examined during a routine postoperative visit at the clinic. The examination included height and body mass measurements, blood pressure, and the resting electrocardiogram (ECG). Then, they underwent a clinical interview to exclude those with heart failure, hepatic or renal insufficiency, infections, coronary artery disease, inflammatory disorders, hormonal replacement therapy, diabetes, and those who supplemented antioxidants 3 months prior to the visit. Patients with a known chronic inflammatory disease (rheumatoid arthritis, systemic lupus erythematosus, Crohn’s disease, Hashimoto’s thyroiditis, psoriasis), recent antibiotic treatment, or intercurrent infections before the replacement surgery, Paget’s disease, revision arthroplasty, vascular disorders (lymphoproliferative disorders, autoimmune hemolytic anemia), or cancer were excluded from the study because the cytokines response can be abnormal in these conditions [27,28]. Eventually, 41 patients after total hip (n = 29, 71%) or knee (n = 12, 29%) replacement, aged 61.0 ± 8.1 years, were enrolled in the study. Overall, 19 were female (46%), and 22 were male (54%). All of them were diagnosed, using the Kellgren Lawrence scale [29], with the stage 4 OA changes in the treated joint, so the study group was homogenic in terms of risk factors for the stage 4 OA (age, sex, and BMI). On the first examination day, the patients were 89.6 ± 9.7 days after the replacement surgery. Five enrolled patients were excluded from the study during its duration due to additional health problems.

### 2.2. General Rehabilitation Program

The patients enrolled in the study started the general rehabilitation protocol ca. 90 days after the endoprosthesis implantation surgery. The sessions took place for 21 consecutive days, starting at 8:00–8:45 a.m. The main parts of the general rehabilitation program were daily physical activity training, physiotherapy, and nutritional workshops. The appointments with a dietitian aimed to teach the patients on the beneficial effects of a healthy diet, and also revise and adjust patients’ individual dietary plans. However, the adjustments were not monitored, so the effect could not be quantified. The importance and the benefit of maintaining the learned behavior for the rest of life were stressed to the patients on every occasion. The physical activity was individually adjusted for each patient, but in general, the program comprised 30–45 min of aerobic walking, 20–30 min of strength training, 30–45 min of bicycle/rotor training, and finally, 15 min of cooldown. Before the sessions started, the patients took the 6-min walk test (6MWT), so the supervising physiotherapist could prepare the appropriate training plan.

### 2.3. Samples Collection

The blood samples for cytokines analysis were collected in the morning, before breakfast at 8:00 a.m., before the first and after the last physiotherapy session. The blood (5 mL) was collected from the ulnar vein to clot activator tubes (S-Monovette, SARSTEDT, Nümbrecht, Germany) and EDTA tubes (1.6 mg/mL EDTA-K3; S-Monovette, SARSTEDT). The samples for serum analysis were centrifuged for 10 min at 4000 rpm at 4 °C. Afterward, the plasma and the serum samples were frozen and stored at −80 °C for further analysis.

### 2.4. Analysis of Cytokines

The cytokine parameters were quantitated using the Bio-Plex 200 System (Bio-Rad Laboratories Inc., Hercules, CA, USA) based on xMAP suspension array technology designed for the multiplexed quantitative measurement of multiple parameters in a single well using 50 µL samples. The immunoassay (The Bio-Plex Pro^TM^ Human, Cytokine 27-plex Assay #M500KCAF0Y; Bio-Rad Laboratories Inc., Hercules, CA, USA) allowed to measure the levels of 27 different pro- and anti-inflammatory parameters: basic fibroblast growth factor (FGF basic), eotaxin-1/CCL11, granulocyte colony-stimulating factor (G-CSF), granulocyte macrophage-colony stimulating factor (GM-CSF), interferon-γ (IFN-γ), macrophage/monocyte chemoattractant protein-1 (MCP-1, MCAF/CCL2), macrophage inflammatory protein-1 alpha (MIP-1α/CCL3), macrophage inflammatory protein beta (MIP-1β/CCL4), interferon-γ induced protein 10 (IP-10/CXCL-10), platelet-derived growth factor-BB (PDGF-BB), RANTES/CCL5, tumor necrosis factor-alpha (TNF-α), vascular endothelial growth factor (VEGF), and interleukins: IL-1β, IL-1RA, IL-2, IL-4, IL-5, IL-6, IL-7, IL-8, IL-9, IL-10, IL-12 (p70), IL-13, IL-15, IL-17A. All procedures were as per the manufacturer’s guidelines. Standard curves for each studied parameter were performed using respective cytokine standard solutions. The serum samples were diluted 4-fold with sample diluent. The standards, controls, and serum samples were incubated with antibody-conjugated beads for 30 min and washed afterward. Next, detection antibodies were added to each well and incubated for 30 min. After removing the unbound detection antibodies with washing buffer, streptavidin-PE was added to each well and incubated for 30 min. The unbound streptavidin-PE was removed using washing buffer. Eventually, the angiogenesis parameters or cytokines bound to beads were analyzed in the Bio-Plex Array Reader. The fluorescence intensity was evaluated using Bio-Plex Manager 6.2 software from the Bio-Plex 200 System. The intra-assay %CV varied from 5 to 15%, and the inter-assay %CV varied from 4 to 11%, depending on the analyzed parameter.

### 2.5. Statistical Analysis

Statistical analysis was performed using data analysis software system Statistica, version 13.3.0 (TIBCO Software Inc., Palo Alto, CA, USA). The distribution of variables was evaluated with the Shapiro–Wilk test and quantile–quantile plot. The interval data with normal distribution were expressed as a mean value ± standard deviation (M ± SD). The interval data with skewed or non-normal distribution were expressed as a median (lower; upper quartile) (Me (Q1;Q3)). The *t*- Student’s test or non-parametric Wilcoxon’s test for dependent variables were used for data comparison. The χ^2^ test was used to assess the relationship between qualitative variables. Statistical significance was set at a *p* < 0.05, and all tests were two-tailed.

## 3. Results

The baseline characteristics of the studied population of patients were published in Skrzep-Poloczek et al. [11]. The cytokine levels were analyzed for 36 patients aged 58.8 ± 8 years (18 women and 18 men), as 5 patients were excluded from the study due to health conditions occurring in the course of the 21-day rehabilitation program.

Biochemical and morphological characteristics of the blood of patients before and after the 21-day general rehabilitation program are also presented in Skrzep-Poloczek et al. [11]. In that paper, we reported that the individually designed general rehabilitation had positive effects on the patients’ blood glucose and lipid concentrations. Glucose, total cholesterol, LDL, and triglycerides levels were significantly lower, and HDL levels were significantly higher when compared to their initial levels, before the rehabilitation started. Furthermore, we observed that CRP level, platelets count, and hematocrit were lower after the rehabilitation, proving that general rehabilitation helped to reduce inflammation and prevent clot formation.

The concentrations of cytokines before and after the 21-day general rehabilitation program are presented in Table 1. The levels of 13 out of the 27 cytokines changed distinctly after the 21-day rehabilitation program. The levels of IL-1RA, IL-2, IL-4, IL-5, IL-6, IL-10, IL-13, IL-15, IL-17A, eotaxin-1/CCL11, MIP-1α/CCL3, RANTES/CCL5, and VEGF increased after the rehabilitation concluded, whereas the level of FGF basic decreased (Table 1).

For the anti-inflammatory factors, we noticed that the level of IL-1RA was higher by 11.76 ± 31.79 pg/mL (95% CI:0.30–23.22), IL-4 by 0.83 ± 1.21 pg/mL (95%CI: 0.41–1.26), IL-10 by 0.45 ± 0.52 pg/mL (95%CI: 0.26–0.63), and IL-13 by 1.05 ± 2.56 pg/mL (95%CI: 0.11–1.99). Moreover, the 21-day rehabilitation program increased the level of IL-6 by 0.37 ± 0.67 pg/mL (95%CI: 0.13–0.62).

In addition, the levels of other inflammatory factors were higher after the 21-day rehabilitation program. The level of IL-2 was 4.21 (2.53;5.23) [pg/mL] compared to 3.78 (2.47;4.78) [pg/mL], and of IL-5 was 11.5 (10.11;17.79) [pg/mL] compared to 10.32 (7.90;15.13) pg/mL] before rehabilitation. For IL-15, it increased by 22.39 ± 35.84 pg/mL (95%CI: 9.24–35.54).

We also noticed that after the 21-day rehabilitation, the levels of some chemokines increased: for RANTES/CCL5 by 749.97 ± 1221.88 pg/mL (95%CI: (316.71–1183.23) (*p* < 0.01), and for MIP-1α/CCL3 by 0.34 ± 0.89 pg/mL (95%CI: 0.03–0.66) (*p* < 0.05).

As for the rest of the studied cytokines, only eotaxin-1/CCL11 and VEGF levels were higher after rehabilitation compared to their levels before rehabilitation: by 40.79 ± 37.17 pg/mL (95%CI: 27.39–54.20) and by 32.64 ± 46.04 pg/mL (95%CI: 16.05–49.25), respectively.

The only statistically significant decrease (*p* < 0.05) in the cytokine level was found for FGF basic. After rehabilitation, the FGF basic level was lower by 3.03 ± 6.53 pg/mL (95%CI: 0.71–5.34). The 21-day general rehabilitation program did not influence pro-inflammatory factors: IFN-γ, TNF-α, and IL-12 (p70), chemokines: IL-8, GM-CSF, and MCP-1/CCL2. Moreover, we did not observe any effect of the 21-day general rehabilitation program on IL-7, IL-9, IL-17, IP-10, G-CSF, MCP-1/CCL2, MIP-1β/CCL4, and PDGF-BB levels (Table 1).

## 4. Discussion

In this study, we assessed the effect of postoperative 21-day general rehabilitation on selected cytokines in patients after hip or knee replacement surgery in the course of osteoarthritis (OA). To the best of our knowledge, this is the first comprehensive study showing the changes in the systemic anti-inflammatory and neuroendocrine-immune regulating factors that may reflect the effectiveness and the clinical benefits of the 21-day general rehabilitation program in patients 90 days after hip or knee implantation.

Since the pro-inflammatory cytokines interleukin-1 beta (IL-1β) and tumor necrosis factor-alpha (TNF-α) play a major role in the inflammatory cascade and cartilage catabolism, their systemic concentrations are abnormally high in OA patients [30,31]. Patients with chronic low-grade inflammation show a higher level of serum TNF-α, IL-1β, and interleukin 6 (IL-6) produced by macrophages derived from adipose tissue [30,31]. These pro-inflammatory cytokines regulate, via autocrine and paracrine mechanisms, the adipocytes’ proliferation, their physiology (enhance lipolysis/inhibit lipid synthesis), and initiate their apoptosis, and also decrease lipids levels in the blood. The elevated levels of TNF-α, IL-1β, and IL-6 have been found in the synovial fluid, synovial membrane, subchondral bone, and cartilage of OA patients, confirming their important role in OA pathogenesis [32,33]. TNF-α, IL-1, and IL-6 play a major role in OA’s cartilage matrix degradation and bone resorption. They can enhance the production of other cytokines, matrix metalloproteinases (MMPs), and prostaglandins, and suppress the synthesis of proteoglycans and type II collagen [34]. On the contrary, Paquet et al. [35] showed that, although in general cytokine levels increased earlier in arthritic knees, compared to contralateral saline-injected knees, some mediators, such as IP-10, IL-12p70, TNF-α, or IL-5, remained unaffected by the arthritic process. No change was also noted for the growth factors, G-CSF and GM-CSF, in all biological fluids, whereas IL-4 and IL-10 levels slightly decreased [35]. Our study showed almost equal systemic baseline levels of inflammatory cytokines, IL-1β, TNF-α, before and after the 21-day general rehabilitation program, which could indicate a stable and controlled inflammatory status of patients after hip or knee implantation, and that the physical activity influenced the inflammatory processes.

However, the 21-day general rehabilitation program increased the level of IL-6. IL-6 is a cytokine that has both pro- and anti-inflammatory properties. It has been frequently investigated in the context of circulatory responses to a single bout of exercise [36]. IL-6 mediates anti-inflammatory effects showing after a single bout of exercise and as a consequence of training adaptation. As a pro-inflammatory factor, IL-6 decreases the production of type II collagen in cartilage and increases the MMPs production [16,37]. Moreover, it is the key cytokine causing changes in the subchondral bone layer [38] and promoting the osteoclasts formation and thus, bone resorption. It shows synergism with other pro-inflammatory factors, IL-1β and TNF-α [39,40]. However, IL-6 is also produced by muscle fibers, via a TNF-independent pathway, during physical activity [41]. IL-6 stimulates other anti-inflammatory cytokines, such as IL-1RA and IL-10, and suppresses the production of the pro-inflammatory cytokine TNF-α [42]. It also enhances lipid turnover, stimulating lipolysis and fat oxidation [43]. Regular physical activity suppresses TNF-α and thereby protects against TNF-α-induced insulin resistance [44,45]. Moreover, IL-6 was introduced as the first myokine, a cytokine produced and released by contracting skeletal muscle fibers, affecting other organs [46] that can be involved in mediating the beneficial effects of exercise and in protecting against chronic diseases associated with low-grade inflammation such as diabetes and cardiovascular diseases [45]. In our study, the IL-6 level was significantly higher after 21 consecutive days of moderate physical activity, confirming the beneficial outcome of the rehabilitation treatment. The anti-inflammatory effects of IL-6 are also confirmed by the IL-6 stimulating effect on IL-1RA and IL-10 production [47], which was also observed after the 21-day rehabilitation program described in our study. The post-exercise increase of IL-10 and IL-1RA in the circulation also mediates the anti-inflammatory effects of exercise. The studies showed that IL-10 inhibits the synthesis of a large spectrum of pro-inflammatory cytokines by different cells, particularly of the monocytic lineage, suggesting that IL-10 acts as an anti-inflammatory factor. IL-10 inhibited the production of IL-1α, IL-1β, and TNF-α and IL-8 and macrophage inflammatory protein-α from LPS-activated human monocytes and other chemokines [45]. IL-10 plays an important role in inducing the inflammatory macrophage/monocyte activation and also inhibits the production of IL-8 in human neutrophils [48]. Although IL-10 influences a broad spectrum of cytokines, IL-1RA inhibits signaling transduction through the IL-1 receptor complex [49]. The IL-1RA belongs to the IL-1 family that binds to IL-1 receptors but does not induce any intracellular response. IL-1RA is also an acute phase protein [50].

Interleukin-2 (IL-2) shows pleiotropic effects. It is required for effective lymphocyte proliferation and differentiation, and it regulates T cell expansion and survival. Targeting IL-2 effect on regulatory T (Treg) cells is used in several immune-related diseases, including chronic graft-versus-host disease (cGVHD), type 1 diabetes (T1D), and systemic lupus erythematosus (SLE) [51]. Cytokines IL-2, IL-7, and IL-15 stimulate the growth of T cells. However, their ratio inducing T cell proliferation remains unknown [52]. Our study showed an increase in IL-2, IL-4, IL-5, IL-13, and IL-15 concentrations in the serum of patients subjected to the 21 daily sessions of general rehabilitation after hip or knee replacement. Studies showed that the circulating levels of IL-15 significantly increased after a single session of resistance exercise in untrained and trained individuals [53]. The IL-15 levels and IL-15 mRNA levels in skeletal muscle significantly increased after eccentric exercises, suggesting that they result from damage to fibers [53], whereas no change in IL-15 levels was found in endurance exercise [54]. The data suggest that IL-15 levels are upregulated in human skeletal muscle after strength training, and the anabolic effects of IL-15 mainly result from a decrease in the proteolytic rate [55]. IL-15 shows a high affinity to the endothelial cells, binds with them, and stimulates angiogenesis. Thus, acute changes in IL-15 levels after exercise may be related to the changes in the blood supply requirements and neovascularization [56].

The analysis of the results showed that the serum levels of eotaxin-1/CCL11, MIP-1α/CCL3, RANTES/CCL5, and VEGF increased after the 21-day general rehabilitation program, when compared to the levels before rehabilitation. It has been reported that the concentrations of some chemokines, like IL-8, GRO-α, RANTES/CCL5, and MCP-1α/CCL3, are higher in OA patients [57]. Hsu et al. [57] reported higher eotaxin-1 and RANTES, and MCP-1α (chemokines) concentrations in the plasma of OA patients. Moreover, they showed that several chemokines, including IL-8, MCP-1/CCL2, MIP-1α/CCL3, MIP-1β/CCL4, and RANTES/CCL5, are overproduced in arthritic joints [57]. Osteoarthritis severity and the degree of the accompanying pain also positively correlate with the increased levels of vascular endothelial growth factor (VEGF) in the synovial fluid [58]. Studies showed that exogenous VEGF stimulates chondrocytes to produce increased levels of pro-inflammatory factors IL-1β, IL-6, and TNF-α [59]. On the other hand, other studies show that both regular and acute physical exercise increase VEGF levels. Sandri et al. [60] and Adams et al. [61] noted elevated VEGF levels after an acute aerobic exercise session in elderly patients with peripheral arterial occlusive disease and ischemic coronary artery disease. Park et al. [62] noted increased VEGF levels after 12 weeks of physical activity, whereas Sandri et al. [63] after 4 weeks of aerobic exercises. Physical activity promotes many alterations in the cardiovascular components: increased blood flow in the exercised area, increased blood shear rate, consecutive increase of nitric oxide levels, systolic volume, tension, and others. Hemodynamic stimuli caused by increasing shear stress and tension on the vessels wall may cause VEGF release. Besides these factors, hypoxic conditions promote VEGF mRNA translation to VEGF, and this process also depends on the physical exercise intensity [64].

Basic fibroblast growth factor (bFGF) shows pleiotropic effects related to mitogenesis, cell migration and differentiation, depending on the affected tissue [65], but mostly it takes part in tissue repair and neovascularization [65,66]. El-Fetiany et al. [65] demonstrated that the FGF basic plasma levels were significantly higher in OA patients than in the control group. Moreover, Honsawek et al. [67] found that levels of FGF basic measured in plasma and synovial fluid of 35 patients with primary knee OA significantly higher comparing with the healthy individuals. They noticed that the plasma levels of FGF basic were related to the OA radiographic severity degree, evaluated by Kellgren Lawrence grading scale [29], and negatively correlated with the cartilage thickness of medial and lateral femoral condyles [67]. In our study, the FGF basic serum level significantly decreased after 21 days of physical activity compared to the initial levels measured in the same patients before rehabilitation, which may prove the beneficial clinical outcome of the applied rehabilitation program.

Our findings are consistent with other studies showing that the increased production of pro-inflammatory and chemoattractive mediators is associated with OA pathology [68]. However, this is the first study showing the increase when comparing levels of the selected markers before and after rehabilitation, in systemic chemokines and cytokines in OA patients who took part in the 21-day general rehabilitation program after hip or knee replacement. The previous study [11] showed increased oxidative processes and oxidative stress levels. The 21-day rehabilitation program started shortly after the surgery increased total superoxide dismutase (SOD) activity and its Cu-Zn isoform (CuZnSOD) activity. The increased levels of pro-inflammatory cytokines, in relation to oxidative stress and other dysregulated mediators, may represent potential therapeutic targets for rehabilitation treatment in OA patients after hip or knee replacement.

The presented study has several limitations. Firstly, it included only 41 patients (19 female and 22 male). Secondly, the study design did not include a control group of healthy individuals but compared the conditions before and after the rehabilitation. The study aimed to observe the effects of the 21-day rehabilitation program on selected parameters in OA patients after hip or knee surgery. Since most patients subjected to hip or knee surgical replacement treated in the outpatient clinic were suffering from additional conditions excluding them from the study, and the project was approaching the deadline, we decided to proceed with the presented experimental setup and collect as much data as possible.

## 5. Conclusions

The presented results indicate that the same levels of pro-inflammatory cytokines (IFN-γ, TNF-α, IL-12 (p70)) before and after the 21-day general rehabilitation reflect a dynamic inflammatory status of osteoarthritis patients after hip or knee implantation. The physical activity increased the levels of inflammatory markers such as RANTES/CCL5, MPI-1α/CCL3, and IL-5 showing a dynamic balance in the pro/anti-inflammatory processes. Significantly higher levels of anti-inflammatory factors after 21 consecutive days of moderate physical activity confirm the beneficial outcome of the applied general rehabilitation program. The increased level of IL-6 after the 21-day rehabilitation program may reflect its anti-inflammatory effect in osteoarthritis patients subjected to prior hip or knee replacement surgery. When it comes to anti-inflammatory cytokines, our results may indicate continuous attempt to restore local cytokine homeostasis after moderate physical activity applied in patients after hip or knee replacement surgery. However, increased levels of numerous pro-inflammatory factors suggest that the mechanisms triggered by the 21-day rehabilitation were insufficient, and alternative rehabilitation protocol should be applied.

## Figures and Tables

**Table 1 biomolecules-12-00605-t001:** The levels of interleukins and cytokines in the serum of patients after hip or knee replacement in the course of osteoarthritis before and after the 21-day general rehabilitation program.

Parameter [pg/mL]	Before Rehabilitation	After Rehabilitation	t/z *	*p*
IL-1β	1.49 ± 1.62	1.59 ± 1.44	1.31	0.199
IL-12 (p70)	3.04 ± 1.61	3.27 ± 1.69	0.66	0.515
IFN-γ	2.20 ± 1.15	1.92 ± 1.11	1.32	0.197
TNF-α	13.47 ± 5.95	14.88 ± 6.16	1.90	0.067
IL-2	3.78 (2.47;4.78)	4.21 (2.53;5.23)	3.84 *	<0.001
IL-6	1.28 ± 0.80	1.64 ± 0.86	3.10	<0.01
IL-1RA	121.51 ± 64.60	133.27 ± 61.21	2.09	<0.05
Il-4	2.65 ± 1.04	3.49 ± 1.31	3.97	<0.001
IL-10	1.89 ± 1.16	2.34 ± 1.16	4.92	<0.001
IL-13	5.60 ± 2.21	6.64 ± 2.68	2.27	<0.05
IL-15	110.86 ± 32.88	133.25 ± 29.71	3.48	<0.01
IL-5	10.32 (7.90;15.13)	11.5 (10.11;17.79)	2.52 *	<0.05
IL-8/GRO	9.13 (7.85;14.27)	9.98 (6.11;12.01)	1.61 *	0.106
GM-CSF	3.27 (2.73;5.09)	3.54 (2.23;5.60)	0.20 *	0.936
MCP-1 (MCAF/CCL2)	30.72 ± 14.06	33.00 ± 13.23	0.83	0.412
MIP-1α/CCL3	2.37 ± 0.91	2.72 ± 1.36	2.21	<0.05
MIP-1β/CCL4	61.83 ± 27.73	57.27 ± 30.28	1.26	0.216
RANTES/CCL5	4893.73 ± 1306.61	5643.70 ± 1617.48	3.53	<0.01
IL-7	6.93 ± 3.95	6.99 ± 3.94	0.07	0.944
IL-9	50.75 ± 18.89	44.29 ± 24.26	1.48	0.150
IL-17A	10.36 ± 3.92	11.02 ± 4.59	0.89	0.381
eotaxin-1/CCL11	112.53 ± 50.03	153.33 ± 53.83	6.21	<0.001
FGF basic	24.80 ± 11.99	21.77 ± 8.98	2.66	<0.05
G-CSF	24.28 ± 14.61	31.04 ± 19.89	1.34	0.189
IP-10/CXCL-10	955.07 ± 438.41	956.14 ± 389.37	0.02	0.987
PDGF-BB	1726.01 ± 1691.87	1817.52 ± 1536.96	0.97	0.341
VEGF	93.16 ± 44.54	125.80 ± 61.33	4.01	<0.001

Legend: FGF basic—basic fibroblast growth factor, G-CSF—granulocyte colony-stimulating factor, GM-CSF—granulocyte-macrophage colony-stimulating factor, GRO—growth-related oncogene, IFN-γ—interferon γ, IL—interleukin, IL-1RA—interleukin-1 receptor antagonist, IP-10/CXCL-10—interferon-γ induced protein 10, MCP-1, MCAF/CCL2 macrophage/monocyte chemoattractant protein-1, MIP-1α/CCL3—macrophage inflammatory protein-1 alpha, MIP-1β/CCL4—macrophage inflammatory protein-1 beta, PDGF-BB—platelet-derived growth factor-BB, TNF-α—tumor necrosis factor-alpha, and VEGF—vascular endothelial growth factor; *—Wicoxon’s test.

## Data Availability

The original data are available after contact with the corresponding author (D.S.).

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
