# Peer review of "The Effects of 21-Day General Rehabilitation after Hip or Knee Surgical Implantation on Plasma Levels of Selected Interleukins, VEGF, TNF-α, PDGF-BB, and Eotaxin-1"

_biomolecules, 2022, doi:10.3390/biom12050605_

Round 1
Reviewer 1 Report
- The manuscript contains typing errors (28, 34, 153, 196, etc.), need correction.
- Please use chemokine names according the new nomenclature (e.g. MIP-1/CCL3 etc.). What type of eotaxin was measured?
- IL-4/5/13 are pleiotropic cytokines, which function depends on the context. They are the signature cytokines of the type II inflammatory response (e.g. allergic inflammation), and are not “pure” anti-inflammatory itself. This also applies to several other cytokines not correctly classified into pro- or anti-inflammatory. Therefore, please reword the sentence on line 26/27 and 84/85, 195/197, 198/200. Moreover, review the main message of the article: rehabilitation (quite short by the way) had clear beneficial anti-inflammatory effect.
- Please specify how many parallel measurements were done for each probes in xMAP?
- Please provide the intra- and inter-assay CV% for analysed biomarkers (Bio-Plex). Have any of the cytokine measured by Bio-Plex tests been validated against ELISA?
- Please specify what Wilcoxon’s test was used (line 181, but also in presenting the results)
- Please correct the presentation of data in Results according the rules stated in statistic: normally distributed M ± SD or non-normal distribution - median (lower; upper quartile). If log-transformed data was normally distributed please show it and present accordingly (M ± SD).
- What means “IL-4 by 0.83 ± 1.21 pg/mL (95%CI: 0.41-1.26)” on lines 195-199? These are not normally distributed data (0.83-1.21 is a negative value). Please explain.
- Please specify what boundaries were used in the manuscript (e.g. on line 202: 4.21 (2.53;5.23) etc.).
- Please add the explanation to Table 1 about what boundaries and what statistical methods and the p-value are presented. Please give exact p-values (not use > or <). Do not use M ± SD for clearly non-normal data in Table 1 and others (e.g. IL1b).
- Please present in “Results” the data and not include into text the elements of discussion (e.g. line 198-200: “no change in levels of pro-inflammatory factors may indicate that the registered change for IL-6 reflects its anti-inflammatory effect”).
- It is not correct to state in Discussion that you demonstrated only anti-inflammatory effect of the rehabilitation. You presented the increase of CCL5/RANTES and MIPs, which are chemotactic (proinflammatory) chemokine, recruiting immune cells to the site of inflammation. That aspect must be discussed.
- You published very nice paper about oxidative stress markers (ref 11). That contained also routine clinical data (CRP etc.). Adding and analysing these data with the current ones would add a lot to the message of the article. Please consider adding at least routine clinical data to the article.
- Please rephrase or remove sentence on lines 308-309. That is confusing in the context of the manuscript. Yes, OA is classified as non-inflammatory to separate from RA, but chronic inflammation is the part of the OA pathogenesis. OA is not a “wear and tear” arthritis.
- OA has some systemic features and the level of several catabolic markers and cytokine concentrations increase step-by-step in higher grades of the disease. In addition, a sex-dependent features of OA pathogenesis have been described. Please provide the data about the radiographic stage of OA before operation. Please analyse if the OA stage, and gender, age or BMI could be an important cofactors for cytokines and chemokines evaluation.
- Please rephrase or remove the sentence on lines 343-344. As you had no healthy controls, you cannot make conclusion about the increased/decreased cytokines in your patients.
- Some authors believe that hip and knee OA are different diseases. Please analyse hip and knee patients’ subgroups separately or use that feature (location) in the statistical analysis as a confounding factor (adjusting your analysis for confounders).
- Please think about how to make better conclusion of your work. Statement that proinflammatory status did not changed, is not correct. You demonstrated the increase of several chemokines and growth factors.
Author Response
Reply to Review Reports
We would like to thank the reviewers for their valuable comments on our manuscript. Each statement has been carefully considered and responded to. All changes are highlighted in blue in the manuscript.
The responses to individual comments are given below.
The authors
Reviewer 1
- The manuscript contains typing errors (28, 34, 153, 196, etc.), need correction.
Thank you for noticing this. Some typing errors, especially the symbols for alpha, gamma and micro letters, resulted from unsuccessful file template conversion between different versions of the editing software. We have corrected all of them and hope that the situation does not repeat this time.
- Please use chemokine names according the new nomenclature (e.g. MIP-1/CCL3 etc.). What type of eotaxin was measured?
Thank you for this suggestion. The new nomenclature has been added in the whole manuscript. In the presented study, we measured eotaxin-1/CCL5 levels, which has been also corrected.
- IL-4/5/13 are pleiotropic cytokines, which function depends on the context. They are the signature cytokines of the type II inflammatory response (e.g. allergic inflammation), and are not “pure” anti-inflammatory itself. This also applies to several other cytokines not correctly classified into pro- or anti-inflammatory. Therefore, please reword the sentence on line 26/27 and 84/85, 195/197, 198/200. Moreover, review the main message of the article: rehabilitation (quite short by the way) had clear beneficial anti-inflammatory effect.
Thank you for this valuable comment.
We decided to include selected IL-4 and IL-13 into the group of anti-inflammatory cytokines based on studies reporting their anti-inflammatory effects in the OA pathophysiology.
The changes have been added to the manuscript: lines 26, 86-99, and 232.
However, after re-reading the manuscript after revision, we would like to keep the main message of the study unchanged.
As for duration of the rehabilitation program, the 21-day period comprises the optimal minimal treatment time in the rehabilitation division of the outpatient clinic, during which the adaptive changes can be observed and the motor functions can be improved. The 21-day rehabilitation comprises a minimal time to benefit from kinesiotherapy.
- Please specify how many parallel measurements were done for each probes in xMAP?
Each sample was measured in triplicate, as suggested in the manufacturer’s guidelines.
- Please provide the intra- and inter-assay CV% for analysed biomarkers (Bio-Plex). Have any of the cytokine measured by Bio-Plex tests been validated against ELISA?
According to the manufacturer’s Bulletin 5828: for the Cytokine 27-plex Assay, the Intra-assay %CV varied from 5 to 15% and the inter-assay %CV varied from 4 to 11%, depending on the analyzed parameter. Since this information is easily accessible on the manufacturer’s website, we resigned from showing it for the individual parameters in the manuscript.
No, the results for cytokines measured by the Cytokine 27-plex Assay were not validated against ELISA.
- Please specify what Wilcoxon’s test was used (line 181, but also in presenting the results).
The t- Student’s test or non-parametric Wilcoxon’s test for dependent variables were used for data comparison. The information has been added to 2.5 section (lines 203-204).
- Please correct the presentation of data in Results according the rules stated in statistic: normally distributed M ± SD or non-normal distribution - median (lower; upper quartile). If log-transformed data was normally distributed please show it and present accordingly (M ± SD).
As is presented in section 2.5 Statistical analysis, the interval data with skewed or non-normal distribution were expressed as a median (lower; upper quartile) (Me (Q1;Q3)), and data were with normal distributions were as mean values ± standard deviation. In the case of variables subjected to logarithmic transformation, the logarithmic transformation ensures that the testing conditions are met, while the results are presented as medians and interquartile range from the variable before the logarithmic transformation. Please note that logarithmic values do not make any clinical sense and therefore are not reported and the results are based on "raw" data.
Differences between groups should be estimated with a point estimate (e.g., absolute differences in means or proportions, odds ratios, or hazard ratios, whichever is the most appropriate for the outcomes) and confidence interval (CI) [1].
Accordingly, the column in Table 1 titled "Parameter" presents the results for individual indicators in accordance with the standard depending on the type of distribution.
Sources:
- https://www.jto.org/article/S1556-0864(20)30679-1/pdf
- What means “IL-4 by 0.83 ± 1.21 pg/mL (95%CI: 0.41-1.26)” on lines 195-199? These are not normally distributed data (0.83-1.21 is a negative value). Please explain.
The differences obtained in the Student's t-test for dependent samples are given in lines 233-238. The differences are given as mean values ± standard deviation, but we cannot interpret these results in the way that the standard deviation is, subtracted from the mean value and then check whether this value is negative. That would be a statistical error. Therefore, for each difference value, we also give the 95% confidence interval. By definition, the confidence interval also takes into account the square root of the sample size, so it is more useful in a clinical interpretation. Thus, when analyzing the confidence intervals, if it does not include the value "0", we say that the differences before and after are statistically significant. When the confidence interval includes the value “0”, no statistically significant differences in the results are stated, and the difference and deviation are not calculated standard.
- Please specify what boundaries were used in the manuscript (e.g. on line 202: 4.21 (2.53;5.23) etc.).
As stated in 2.5 Statistical analysis section, the interval data with skewed or non-normal distribution were expressed as a median (lower; upper quartile) (Me (Q1;Q3)). However in the previous version of the text in the section the symbol between Q1 and Q3 in the brackets was incorrect (“-“ instead “;”) and in has been corrected in the current version.
- Please add the explanation to Table 1 about what boundaries and what statistical methods and the p-value are presented. Please give exact p-values (not use > or <). Do not use M ± SD for clearly non-normal data in Table 1 and others (e.g. IL1b).
In clinical analysis, the p-value reporting standard assumes that for statistically insignificant results we give the exact value, while for significant results we give the value in the form of ranges < 0.05, < 0.01, and < 0.001. In the presented results, p < 0.001 appears quite often. Since we cannot give the value 0.00000, because we calculated the probability that will never reach such a value, it is recommended by the international statistical societies to recognize an error of 0.1% (written as <0.001). Therefore, we decided to not present exact p-values in Table 1 for statistically significant results. According to the statistical recommendations ‘not significant’ (‘NS’) for p-value ≥ 0.05 should not be reported either but, in this case, we decided to do otherwise.
Source: https://bjui-journals.onlinelibrary.wiley.com/doi/full/10.1111/bju.14640
- Please present in “Results” the data and not include into text the elements of discussion (e.g. line 198-200: “no change in levels of pro-inflammatory factors may indicate that the registered change for IL-6 reflects its anti-inflammatory effect”).
Thank you for this suggestion. The mentioned phrase has been removed.
- It is not correct to state in Discussion that you demonstrated only anti-inflammatory effect of the rehabilitation. You presented the increase of CCL5/RANTES and MIPs, which are chemotactic (proinflammatory) chemokine, recruiting immune cells to the site of inflammation. That aspect must be discussed.
Thank you for this comment. The changes have been introduced in the Conclusions (lines: 396-399 and 403-408).
- You published very nice paper about oxidative stress markers (ref 11). That contained also routine clinical data (CRP etc.). Adding and analysing these data with the current ones would add a lot to the message of the article. Please consider adding at least routine clinical data to the article.
The information on blood’s biochemical and morphological parameters of the study group has been added in the Results section (lines 212-220).
- Please rephrase or remove sentence on lines 308-309. That is confusing in the context of the manuscript. Yes, OA is classified as non-inflammatory to separate from RA, but chronic inflammation is the part of the OA pathogenesis. OA is not a “wear and tear” arthritis.
The mentioned sentence has been removed as suggested.
- OA has some systemic features and the level of several catabolic markers and cytokine concentrations increase step-by-step in higher grades of the disease. In addition, a sex-dependent features of OA pathogenesis have been described. Please provide the data about the radiographic stage of OA before operation.
All patients included in the study were diagnosed, using the Kellgren Lawrence scale, with the stage 4 OA changes in the treated joint. The studied group was homogenic in terms of risk factors (age, sex, and BMI) for the OA stage 4. This information has been added to the section 2.2 Study group (lines 145-147).
Please analyse if the OA stage, and gender, age or BMI could be an important cofactors for cytokines and chemokines evaluation.
We are grateful for this suggestion. Nevertheless, too few patients were included in the study to perform this kind of analyses.
- Please rephrase or remove the sentence on lines 343-344. As you had no healthy controls, you cannot make conclusion about the increased/decreased cytokines in your patients.
The text in the Discussion section has been changed accordingly, and the experimental setup of the presented study is mentioned to avoid confusion (lines 371 and 375/376).
- Some authors believe that hip and knee OA are different diseases. Please analyse hip and knee patients’ subgroups separately or use that feature (location) in the statistical analysis as a confounding factor (adjusting your analysis for confounders).
We are grateful for this suggestion. Nevertheless, too few patients were included in the study to perform this kind of analyses.
- Please think about how to make better conclusion of your work. Statement that proinflammatory status did not changed, is not correct. You demonstrated the increase of several chemokines and growth factors.
The Conclusions section has been corrected.

Reviewer 2 Report
The effects of 21-day general rehabilitation after hip or knee 2 surgical implantation on plasma levels of selected interleukins, 3 VEGF, TNF-a, PDGF-BB, and eotaxin
By Idzik et al.,
The study investigates the cytokine profile in serum of patients (90 days after joint replacement due to OA) who underwent a 3 week lasting rehabilitation program. It is interesting, matches with the focus of the journal and is in line with literature. Why was the time point of 21 days selected? Was the blood indeed from each patient at 90 days post surgery selected?
Line 228: this selection could be discussed. The limitations of the study should be mentioned. Concerning the blood sample collection: no base line values before TEP surgery are evailable because no samples were collected before surgery. Perhaps available literature concerning this could be discussed. No control was included which means blood from patients which received no physiotherapy post surgery. Hence, we can not conclude that the changes are indeed caused by pühysiotherapy…
Line 28: eotaxin, please explain somewhere what is it (could be done in the discussion, e.g. line 307)?
Last sentence of the abstract concerning IL-6, IL-6 should also be mentioned in lines 26-29
Line 76/99: „IL-6“ perhaps the authors could name it „immunoregulatory“, compare lines 98-99.
Line 78: „autoinflammatory“ plase explain, is OA indeed an autoinflammatory disease? the term might be more approapriate for RA. The term should be explained.
Line 87: „produced by activated macrophages“ it is well known that synovial fibroblasts and even chondrocytes are also activated in OA to produce proinflammatory cytokines
Line 88: „IL-10 upregulates endogenous anti-cytokines“ please provide examples of these mediators and a reference is lacking at the end of the sentence.
Line 109: which negative aspects?
Line 112: „hip and knee“ could differences be detected between both joints?
Line 135: „nutritional workshops“, was the nutrition of the patients changed? Could it influence the systemic cytokine levels.
Line 153: 50 l (µL?)
Line 196: 11.76 something is lacking, „95%CI“ insert consistently a blank or not.
Line 261: „type II collagen“ insert „in cartilage“
Line 308/315/356/361: „osteoarthritis“ the abbreviation „OA“ should be consistently be used
Author Response
Reviewer 2
The study investigates the cytokine profile in serum of patients (90 days after joint replacement due to OA) who underwent a 3 week lasting rehabilitation program. It is interesting, matches with the focus of the journal and is in line with literature.
Why was the time point of 21 days selected?
The 21-day period comprises the optimal minimal treatment time in the rehabilitation division of the outpatient clinic, during which the adaptive changes can be observed and the motor functions can be improved. The 21-day rehabilitation comprises a minimal time to benefit from kinesiotherapy.
Was the blood indeed from each patient at 90 days post surgery selected?
As stated in the 2.2 Study group section, the patients were 89.6 ± 9.7 days after the replacement surgery, on the first examination day. It was impossible to adhere to the 90-day post-surgery rule due to scheduling challenges related to the working hours of the outpatient clinic and unexpected problems on the patients’ side.
Line 228: this selection could be discussed. The limitations of the study should be mentioned.
The paragraph describing the limitations of the study has been added to Discussion section (lines: 348-392).
Concerning the blood sample collection: no base line values before TEP surgery are evailable because no samples were collected before surgery. Perhaps available literature concerning this could be discussed. No control was included which means blood from patients which received no physiotherapy post surgery. Hence, we can not conclude that the changes are indeed caused by pühysiotherapy.
Thank you for this comment. We are aware of this limitation. However, the study aimed to compare the levels of selected parameters in patients before and after rehabilitation program, not in relation to the surgery itself. Therefore, we think that the presented approach is sufficient to explain the observed changes.
Line 28: eotaxin, please explain somewhere what is it (could be done in the discussion, e.g. line 307)?
Eotaxin-1 has been mentioned in the Introduction and classified accordingly (lines: 72-73).
Last sentence of the abstract concerning IL-6, IL-6 should also be mentioned in lines 26-29
The suggested change has been applied (line 27).
Line 76/99: „IL-6“ perhaps the authors could name it „immunoregulatory“, compare lines 98-99.
Thank you for this suggestion. In line 77, we added “and immunoregulatory”. In lines 114-115, using “either/or” is self-explanatory in this context.
Line 78: „autoinflammatory“ plase explain, is OA indeed an autoinflammatory disease? the term might be more approapriate for RA. The term should be explained.
The lines 77-80 (“Pro-inflammatory and immunoregulatory interleukins (IL-1, IL-6, tumor necrosis factor-alpha (TNF-α), interferon gamma (IFN-γ), and granulocyte-macrophage colony-stimulating factor (GM-CSF) play a significant role in the pathogenesis of many autoinflammatory diseases.“) describe the general scope of action of the selected markers, which include actions observed also in
autoinflammatory diseases such as RA. That is why the term “autoinflammatory” is used in the sentence.
Line 87: „produced by activated macrophages“ it is well known that synovial fibroblasts and even chondrocytes are also activated in OA to produce proinflammatory cytokines
Thank you for this suggestion. The additional information has been added to the mentioned sentence (line 102).
Line 88: „IL-10 upregulates endogenous anti-cytokines“ please provide examples of these mediators and a reference is lacking at the end of the sentence.
The sentence has been corrected and the reference has been added. Please see lines 102-104.
Line 109: which negative aspects?
The possible negative aspects of the study were tiredness, additional joint and muscle pain, and time obligation. The sentence has been corrected (lines 125-126).
Line 112: „hip and knee“ could differences be detected between both joints?
The study aimed to showcase the beneficial effects of physical activity and improved motor functions on selected parameters. We did not intend to differentiate between these two types of surgery, as the studied group of patients was too small. However, we appreciate the idea and find it worthy of further investigation.
Line 135: „nutritional workshops“, was the nutrition of the patients changed? Could it influence the systemic cytokine levels.
The 21-day rehabilitation program included appointments with a dietitian in order to teach the patients on the beneficial effects of the healthy diet, and also revise and adjust patient’s individual dietary plans. However, the adjustments were not monitored, so the effect could not be quantified. We have expanded section 2.3, and included this information (lines: 154-157).
Line 153: 50 l (μL?)
Thank you for noticing this typing error. It resulted from unsuccessful file template conversion between different versions of the editing software. We have corrected it and hope that the situation does not repeat this time.
Line 196: 11.76 something is lacking, „95%CI“ insert consistently a blank or not.
Thank you for noticing this. The sentence has been corrected (line: 236-240).
Line 261: „type II collagen“ insert „in cartilage“
The sentence has been corrected as suggested (line 292).
Line 308/315/356/361: „osteoarthritis“ the abbreviation „OA“ should be consistently be used
Thank you for this suggestion. However, we kindly disagree on using OA abbreviations in all these mentioned instances. In the first two instances (former line 308 has been deleted, line 345), OA abbreviation cannot be used as scientific writing guidelines say that one cannot start sentence with an abbreviation. In the last two instances (lines: 396, 402) we purposefully used the full-length term as this section is often read separately from the main text and we wanted to avoid introducing too many abbreviations.

Round 2
Reviewer 2 Report
The authors adressed and discussed my previous comments, changing the manuscript accordingly or explaining why some issues were not changed.
Particularly, the limitations have been clearly stated
It has been improved and therefore, I believe it suitable for publication.